# Genetic and Clinical Characteristics of Korean Chronic Lymphocytic Leukemia Patients with High Frequencies of *MYD88* Mutations

**DOI:** 10.3390/ijms24043177

**Published:** 2023-02-06

**Authors:** Ari Ahn, Hoon Seok Kim, Tong-Yoon Kim, Jong-Mi Lee, Dain Kang, Haein Yu, Chae Yeon Lee, Yonggoo Kim, Ki-Seong Eom, Myungshin Kim

**Affiliations:** 1Department of Laboratory Medicine, Seoul St. Mary’s Hospital, College of Medicine, The Catholic University of Korea, Seoul 06591, Republic of Korea; 2Catholic Genetic Laboratory Center, Seoul St. Mary’s Hospital, College of Medicine, The Catholic University of Korea, Seoul 06591, Republic of Korea; 3Department of Hematology, Catholic Hematology Hospital, Seoul St. Mary’s Hospital, College of Medicine, The Catholic University of Korea, Seoul 06591, Republic of Korea

**Keywords:** chronic lymphocytic leukemia, *MYD88*, *IGHV*, somatic hypermutation, Korea

## Abstract

Chronic lymphocytic leukemia (CLL) is the most common adult leukemia in Western countries. However, CLL is relatively rare in Asia; its genetic features are rarely studied. Here, we aimed to genetically characterize Korean CLL patients and to elucidate the genetic and clinical associations based on data obtained from 113 patients at a single Korean institute. We used next-generation sequencing to explore the multi-gene mutational data and immunoglobulin heavy chain variable gene clonality with somatic hypermutation (SHM). *MYD88* (28.3%), including L265P (11.5%) and V217F (13.3%), was the most frequently mutated gene, followed by *KMT2D* (6.2%), *NOTCH1* (5.3%), *SF3B1* (5.3%), and *TP53* (4.4%). *MYD88*-mutated CLL was characterized by SHM and atypical immunophenotype with fewer cytogenetic abnormalities. The 5-year time to treatment (TTT) of the overall cohort was 49.8% ± 8.2% (mean ± standard deviation) and the 5-year overall survival was 86.2% ± 5.8%. Patients with SHM, isolated del(13q), *TP53*-wild type, and *NOTCH1*-wild type showed better results than those without these conditions. In the subgroup analyses, patients with SHM and L265P presented shorter TTT than patients with SHM but not L265P. In contrast, V217F was associated with a higher SHM percentage and showed a favorable prognosis. Our study revealed the distinct characteristics of Korean CLL patients with high frequencies of *MYD88* mutations and their clinical relevance.

## 1. Introduction

Chronic lymphocytic leukemia (CLL) is characterized by clonal proliferation and accumulation of cluster of differentiation (CD)5-positive mature B cells in peripheral blood (PB), the bone marrow (BM), lymph nodes, and the spleen [1,2]. The diagnosis of CLL requires the presence of at least 5 × 10^9^/L monoclonal B cell counts (5000/μL) for more than 3 months in PB [2]. CLL is clinically and biologically heterogeneous [3]; thus, the clonality of circulating B cells needs to be confirmed using flow cytometry [4,5]. Previous studies have indicated that CLL is also genetically heterogeneous, and some genetic features have been implicated in prognosis and treatment response [6,7,8,9]. These include somatic hypermutation (SHM) of the immunoglobulin heavy chain variable gene (*IGHV*), molecular and cytogenetic aberrations, and several protein expressions in CLL cells [10,11,12,13,14]. Next-generation sequencing (NGS) has enabled efficient identification of these genetic features and is widely used in clinical laboratories. Regarding *IGHV*, clonality and SHM can be analyzed simultaneously using NGS [15]. In addition, mutations in major genes, such as *TP53*, *ATM*, *NOTCH1*, *SF3B1*, and *MYD88*, can be identified conveniently using NGS [13,16,17,18,19].

Although CLL is the most frequent adult leukemia in Western countries, its incidence in Asian populations has been relatively low (age-adjusted incidence rate per 100,000 individuals: UK 6.60, US 6.90, Taiwan 0.39, and Korea 0.20) [20,21,22,23]. Studies comparing Asian and other ethnic populations have demonstrated the rarity of its incidence and distinct phenotypes [22,24,25,26,27,28]. Genetic features in Asian patients with CLL are rarely studied, and their differences from those in other ethnic populations remain unclear.

In this study, we identified the genetic characteristics of patients with CLL admitted to a single Korean institute. We generated mutational data and *IGHV* clonality with SHM using NGS and integrated them with significant clinical and laboratory features in CLL. These genetic characteristics were carefully compared with those for other populations to explain the genotype–phenotype relationships in Asian patients with CLL and reveal the effects of these particular genetic characteristics on the prognosis of patients.

## 2. Results

### 2.1. Clinical and Phenotypic Characteristics

The median patient age at diagnosis was 59 (range: 32–87) years, and the population was predominantly male, with a male-to-female ratio of 2:1 (Table 1). Eleven patients (9.7%) were in Rai stages 3–4 and Binet stage C. Mean white blood cell count and clonal CLL-like cell count were 34.60 × 10^9^/L ± 46.340 × 10^9^/L and 27.52 × 10^9^/L ± 40.281 × 10^9^/L, respectively. As observed using flow cytometric immunophenotyping, all patients were positive (113/113) for CD19, 93.8% were positive (102/113) or dim positive (4/113) for CD5, 94.6% were positive (99/112) or dim positive (7/112) for CD23, and 45.8% were positive (49/107) for FMC7 (Figure 1A).

### 2.2. Genetic Profile of CLL

Fifty-seven percent of the patients (61/107) demonstrated cytogenetic abnormalities, as detected using conventional karyotyping and/or fluorescence in situ hybridization (FISH). Conventional karyotyping showed 48.8% of patients (39/80) with at least one chromosomal abnormality and 33.3% of them (13/39) had complex karyotypes comprising three or more chromosomal abnormalities. Among them, trisomy 12 was the most common aberration (10/39), followed by del(13q) (8/39), del(11q) (7/39), and del(6q) (5/39). FISH revealed abnormalities in 49.5% of the patients (52/105): del(13q) in 39.7% (23/58), isolated del(13q) in 22.4% (13/58), trisomy 12 in 32.0% (16/50), del(6q) in 12.8% (6/47), del(11q) in 10.8% (11/102), and del(17p) in 5.0% (5/101) (Figure 1A).

The NGS-based multi-gene mutation analysis revealed 67 pathogenic or likely pathogenic mutations in 52.2% of patients (59/113) (Appendix A). The median number of mutations was 1 per patient (range: 1–4). *MYD88* (28.3%) was the most frequently mutated gene, followed by *KMT2D* (6.2%), *NOTCH1* (5.3%), *SF3B1* (5.3%), and *TP53* (4.4%) (Figure 1B). Overall, 77.0% of patients showed at least one genetic aberration, including cytogenetic and molecular genetic abnormalities. Trisomy 12 was commonly accompanied by *NOTCH1* (*p* = 0.033), and *ATM* abnormalities were frequently accompanied by *SF3B1* (*p* = 0.016) and del(6q) (*p* = 0.005) (Figure 1C).

Clonal *IGH* rearrangement was identified in 99.1% of patients with CLL (112/113). In V and J repertoires, V4-34_J4 was predominantly detected in 8.9% of cases, followed by V3-23_J4 (6.3%), V3-7_J4 (3.6%), V1-3_J4 (3.6%), and V2-5_J4 (3.6%). The SHM status was identified in 86.7% (104/112), except in eight inconclusive cases (a median percentage of 5.8% [range: 2.03–12.84%]). Patients without SHM (13/104) showed cytogenetic and molecular abnormalities more frequently than those with SHM (91/104) (85.7% vs. 55.3% respectively, *p* = 0.040), including del(6q) (50.0% vs. 8.1%, *p* = 0.027) and del(11q) (33.3% vs. 8.5%, *p* = 0.032), and *NOTCH1* mutation (28.6% vs. 1.1%, *p* = 0.001).

### 2.3. Characteristics of the MYD88 Mutations in CLL

We further investigated the *MYD88* mutations in CLL because *MYD88* was the most frequently mutated gene in our cohort (28.3%, *n* = 32). Among them, the V217F and L265P mutations constituted 47% (*n* = 15) and 41% (*n* = 13), respectively (Figure 2A). The M232T and S219C mutations were detected in three and two patients, respectively. Most *MYD88* mutations were mutually exclusive except one with both S219C and V217F mutations (Figure 1A). The *MYD88* mutations were associated with atypical immunophenotypes (*p* < 0.001), including CD5(−/dim+) (*p* = 0.002), CD23(−/dim+) (*p* = 0.043), and FMC7(+) (*p* = 0.002). Each mutation was associated with a specific atypical immunophenotype: CD5(−/dim+) with L265P mutation (*p* = 0.003), and CD23(−/dim+) and FMC7(+) with V217F (*p* = 0.011, and *p* = 0.020, respectively). The *MYD88*-mutated group showed cytogenetic abnormalities less frequently than the wild type group (*p* = 0.010). All patients with the *MYD88* mutation showed SHM of *IGHV*, whereas only 81.6% of *MYD88*-wild type showed this feature (*p* = 0.010) (Table 1). Patients with L265P revealed lower median SHM% than those with V217F (5.7% [range: 3.19–8.11%] vs. 7.5% [range: 4.44–13.22%], *p* = 0.002) (Figure 2B).

### 2.4. Treatment Outcomes

With a median follow-up of 30.0 months, the 5-year time to treatment (TTT) of the overall cohort was 49.8% ± 8.2% (mean ± standard deviation), and the 5-year overall survival (OS) was 86.2% ± 5.8%. In the survival analysis using the Kaplan–Meier method, long TTT was associated with SHM, isolated del(13q), and *TP53*-wild type (*p* = 0.020, *p* = 0.037, and *p* = 0.029, respectively) (Figure 3A), and long OS was associated with SHM and *TP53*-wild type (*p* = 0.002 and *p* = 0.002, respectively) (Figure 3B). In the univariate analysis, *TP53* abnormality and *NOTCH1* mutation were associated with a poor prognosis in terms of both TTT and OS. In the multivariate analysis, both *TP53* abnormality and *NOTCH1* mutation retained prognostic significance in terms of TTT (*p* = 0.035 and *p* < 0.001, respectively), whereas *TP53* abnormality did so for OS (*p* = 0.015) (Appendix A).

Regarding *MYD88* mutations, patients with L265P tended to show shorter TTT than those with V217F (*p* = 0.108) (Figure 4). In the subgroup analysis of *MYD88* mutations and SHM status, SHM(+)L265P(+) patients tended to show shorter TTT than SHM(+)L265P(−) patients (*p* = 0.098), and SHM(+)L265P(−) patients had longer TTT and OS than SHM(−) patients (*p* = 0.011 and *p* = 0.002, respectively) (Figure 5A). In contrast, SHM(+)V217F(+) patients presented longer TTT and OS than SHM(−) patients (*p* = 0.017 and *p* = 0.020, respectively) (Figure 5B).

## 3. Discussion

CLL is the most common type of leukemia in patients of European descent, whereas its incidence is lower in Asians [20,21]. Asian patients with CLL are younger, with an advanced stage and a higher frequency of atypical CLL than their European counterparts [22,24,25,26,27,29,30]. Studies have ascribed these differences to genetic influences rather than geographic factors [23,26]. However, only a few studies have analyzed the comprehensive genetic data of Asian CLL [30,31,32]. In this study, we investigated the genetic profiles of Korean patients with CLL to identify their characteristics.

A total of 57.0% of patients showed cytogenetic abnormalities—trisomy 12 and del(13q) were the most common abnormalities (33%) in our cohort. These results are in line with those of previous studies [10,27,33,34]. Regarding the SHM of *IGHV*, in our study subjects, 86.7% of SHM involved V4-34, V3-23, V3-7, V1-3, and V2-5. Marinelli et al. [35] demonstrated a similar proportion of involvement in Chinese CLL, with 66% of SHM involving V4-34, V3-23, and V3-7, whereas Italian CLL showed 49% SHM involving V1-69, V4-34, and V3-23. We compared cytogenetic abnormalities to the *IGHV* data and found that the SHM status was associated with cytogenetic risk factors in CLL. High-risk abnormalities, including del(6q) and del(11q) abnormalities, were commonly observed in patients without SHM, as observed in other studies [12,36].

Previous studies have uncovered the genomic landscape of CLL and identified several genes carrying mutations at ˃5% frequency, such as *TP53* (5–15%), *ATM* (9–15%), *MYD88* (2–13%), *SF3B1* (8–21%), and *NOTCH1* (4–13%) [6,13,16,17,18,37,38]. Similarly, using NGS, we observed that 52.2% of patients had one or more pathogenic or likely pathogenic mutations. To compare mutation frequencies between populations, we referred to International Cancer Genome Consortium (ICGC), Dana-Farber data and a recent Chinese study result [18,30]. The most notable difference between Korean and western population was a higher incidence of *MYD88* (28.3% vs. 3.0–3.6%) and *KMT2D* (6.2% vs. 0.7–1.1%) mutations and a relatively lower incidence of *SF3B1* (5.3% vs. 8.6–21.0%) mutation (Figure 1B) [18,37]. In particular, the recent study showed a high incidence of *MYD88* (13%) and *KMT2D* (11%) mutations in Chinese patients with CLL [30], while a considerably higher incidence of *MYD88* mutations (28%) was observed in our cohort of Korean CLL patients. Four recurrent mutations were identified on the Toll/interleukin-1 receptor (TIR) domain (V217F, L265P, M232T and S219C), and the two hotspot mutations, i.e., L265P and V217F, constituted 88% of all *MYD88* mutations.

The *MYD88* mutations promote B cell proliferation and survival via the activation of NF-κB and related signaling pathways [39]. Among these mutations, L265P is frequently observed in Waldenström macroglobulinemia and activated B cell (ABC)-type diffuse large B cell lymphoma (DLBCL) [39,40]. Similarly, in CLL, the *MYD88* mutation is considered an early clonal event and driver mutation. However, research on the biological and clinical relevance of the *MYD88* mutation is in progress [41,42]. *MYD88* L265P is observed in a distinct group of younger patients with CLL with a favorable prognosis [43]. However, other studies have found that *MYD88* mutations are associated with an unfavorable prognosis in CLL patients with SHM [44,45]. Other previous studies have claimed no significant effect of *MYD88* mutation on prognosis [46,47,48]. Therefore, this prognostic relevance is still controversial. The other hotspot mutation, V217F, was present in CLL and DLBCL from Catalog of Somatic Mutations in Cancer (COSMIC) [49]. Although non-L265P, including V217F, showed favorable evolution owing to a lower NF-κB activation capacity in DLBCL [50], the characteristics of V217F were not well defined in CLL.

Interestingly, our data suggested an overall unfavorable effect of L265P and a favorable effect of V217F on CLL prognosis. L265P tended to show a poorer outcome than V217F, which may stem from a lower SHM% for L265P than for V217F. In the subgroup analyses, we found that patients with SHM and L265P had a worse outcome than those with only SHM, and patients with SHM and V217F had better results than those without SHM. B cell receptor (BCR) signaling was more active in CLL without SHM and associated with an aggressive clinical course [51]. In contrast, patients with SHM had anergic BCR and an indolent clinical course [51]. However, in CLL with both SHM and L265P, the latter seems to counteract the effects of SHM. In this study, the prognosis of patients with SHM and L265P was comparable to that of patients without SHM. This result confirmed the findings of previous studies revealing the prognostic significance of L265P in CLL [44,45]. In addition, we found that patients with the *MYD88* mutation had distinctive immunophenotypic, cytogenetic, and molecular features. The *MYD88* mutation was associated with atypical immunophenotypes, including CD5(−/dim+), CD23(−/dim+), and/or FMC7(+), and a higher incidence of SHM of *IGHV*. Moreover, each hotspot mutation was related to a specific immunophenotype: L265P with CD5(−/dim+) and V217F with CD23(−/dim+) and FMC7(+).

The *MYD88* mutations have been identified in several other human malignancies [39,52]. Several therapeutic agents that directly or indirectly inhibit the *MYD88* L265P oncogenic activity in CLL have emerged, such as BTK inhibitors, IRAK4 inhibitors [45], and immunomodulatory oligonucleotides [6,53]. The US Food and Drug Administration has approved the BTK inhibitor ibrutinib for CLL treatment, and acalabrutinib has achieved an overall response rate of 95% in the treatment of relapsed CLL [54]. Improgo et al. (2019) [45] revealed that *MYD88* L265P is associated with a prognostic gene expression signature, and demonstrated the potential of the *MYD88* pathway as a therapeutic target in CLL through IRAK4 inhibition.

Among the other genes, *TP53* and *NOTCH1* were associated with poor outcomes and unmutated *IGHV* in CLL [55,56,57,58,59]. Both genes were determined as poor prognostic factors in this study based on the results of the multivariate analysis, although their mutational incidence was relatively low. We classified patients with *TP53* abnormalities as a high-risk group and started treatment early; their OS and TTT were significantly worse.

In summary, we investigated the genotypic, phenotypic, and clinical characteristics of CLL in a single Korean institute. We observed high frequencies of *MYD88* mutations in Korean patients with CLL compared with that in populations of predominately European descent. Patients with the *MYD88* mutation showed distinctive features, including SHM, atypical immunophenotype, and fewer cytogenetic abnormalities. We evaluated Asian CLL patients with abundant *MYD88* mutations and determined the significance of each *MYD88* mutation type. L265P shows an unfavorable prognosis, and V217F indicates a favorable prognosis within the CLL group with SHM. There were some limitations in our study. Owing to the short follow-up period (approximately 30.0 months) and the small number of patients per subtype, it was difficult to clarify the prognostic significance of all variant types. In addition, despite our best efforts, cytogenetic results were available in 40–87% of patients. Therefore, the characteristics of Korean patients with CLL identified in this study require further validation in a large number of patients, possibly a larger Asian cohort with better data accumulation.

## 4. Materials and Methods

### 4.1. Patients

In total, 113 patients diagnosed with naïve CLL at Seoul St. Mary’s Hospital, Catholic University of Korea, from March 2018 to December 2021 were included in the study. Patients were diagnosed according to the World Health Organization (WHO) Classification of Tumors of Hematopoietic and Lymphoid Tissues on the basis of their clinical features, laboratory findings (PB and/or BM morphology, and flow cytometric immunophenotyping), cytogenetics, and molecular genetics [1]. Morphologic characteristics of CLL were reviewed on PB smears and/or BM aspiration smears and hematoxylin and eosin-stained tissue sections and confirmed independently by two experienced hematopathologists to exclude other small B-cell lymphomas. CLL with atypical immunophenotype was regarded when either CD5 or CD23 were negative or dim positive (−/dim+), or when FMC7 is positive [60]. Extramedullary involvement and localization were assessed using positron emission tomography/computed tomography. Initial staging was performed according to the Rai and Binet systems [61,62]. This study was approved by the institutional review board of Seoul St. Mary’s Hospital (KC21RISI0569).

Decision-making for treatment initiation was performed following the international workshop on CLL 2018 (iwCLL 2018) [63]. Most patients were monitored every 4–6 months without intervention because they did not exhibit any symptoms and signs of advanced disease or evidence of progressive disease (*n* = 73). The combination of fludarabine, cyclophosphamide, and rituximab (FCR) was the most commonly used treatment regimen (*n* = 22), followed by rituximab plus bendamustine (*n* = 6) and obinutuzumab plus chlorambucil (*n* = 5). Other regimens included acalabrutinib monotherapy (*n* = 4), acalabrutinib, venetoclax plus obinutuzumab (*n* = 1), venetoclax plus obinutuzumab (*n* = 1), and venetoclax plus acalabrutinib (*n* = 1).

### 4.2. Flow Cytometric Immunophenotyping

Fresh EDTA anti-coagulated PB or BM aspirates were collected from all patients. Lymphocytes were isolated and analyzed using flow cytometry with a FACSCanto II Flow Cytometer and FACSDiva software (Becton–Dickinson, San Jose, CA, USA). Eighteen monoclonal antibodies were used, namely those against CD2, CD3, CD4, CD5, CD7, CD8, CD10, CD14, CD19, CD20, CD23, CD43, CD45, CD56, CD103, FMC7, surface kappa, and lambda light chains (Becton–Dickinson). A marker was considered as ‘dim positive’ for a uniformly positive population with relative lower fluorescence intensity than a positive normal cell population [60].

### 4.3. Conventional Cytogenetics and FISH

PB or BM aspirates from 80 patients were cultured under stimulated culture conditions and harvested using routine laboratory protocol. Conventional karyotype analysis was performed with G-banded metaphase cells obtained after 72 h of lipopolysaccharide stimulation of the BM aspirates. In each case, at least 20 metaphase cells were assessed, and the cytogenetic abnormalities were interpreted following the 2016 International System for Human Cytogenetic Nomenclature guidelines [64].

FISH analysis was performed on 105 patients following the manufacturer’s instructions. Anomalies were detected using five probes: MYB Deletion Probe for del(6q), ATM Deletion Probe for del(11q), Chromosome 12 Alpha Satellite Probe for trisomy 12, RB1 Deletion Probe for del(13q), and TP53 Deletion Probe for del(17p) (Cytocell, Banbury, UK). Slides were prepared using cells harvested for conventional cytogenetic studies and processed for FISH. A total of 400 interphase or metaphase nuclei were examined in each case.

### 4.4. NGS-Based IGH Gene Rearrangements and IGHV Mutation Analysis

We identified *IGH* rearrangements and assessed the extent of SHM in *IGHV* using the LymphoTrack^®^ *IGHV* Leader Somatic Hypermutation Assay and *IGH* FR1 Assay kits (InVivoScribe, San Diego, CA, USA) in all patients. *IGHV* sequences with <98% similarity with the germline sequence (V gene mutation ≥ 2%) were classified as mutated *IGHV* (SHM), whereas those with ≥98% similarity with the germline sequence were classified (V gene mutation < 2%) as unmutated *IGHV,* and unproductive sequences including out-of-frame junction and/or stop codon were classified as inconclusive *IGHV*, according to the European Research Initiative on CLL (ERIC) recommendation [65].

### 4.5. NGS-Based Multi-Gene Mutation Analysis

Samples of all patients were subjected to NGS using St. Mary’s (SM) customized NGS panel for CLL/lymphoma, named the “SM CLL/lymphoma panel” Ion AmpliSeq Technology (Thermo Fisher Scientific, Waltham, MA, USA). The SM CLL/lymphoma panel consists of 43 genes (Appendix A), an Ion Chef™ system (Thermo Fisher Scientific) and an Ion S5 XL Sequencer (Thermo Fisher Scientific) were used. The sequenced reads were mapped to the human reference genome (hg19, Genome Reference Consortium, February 2009). Annotated variants were classified into four tiers according to the Standards and Guidelines of the Association for Molecular Pathology [66]. Bioinformatic analysis was carried out using both customized and manufacturer-provided pipelines. Variants were selected and annotated using analytic algorithms and public databases. All mutations were manually verified using Integrative Genomic Viewer [67].

### 4.6. Statistical Analysis

All statistical analyses were performed using SPSS for Windows (version 24.0; IBM Corporation, Armonk, NY, USA). Fisher’s exact and chi-square tests were used to determine the correlations among *MYD88* mutations, SHM, and clinical characteristics. In addition, Mann–Whitney U test was applied to compare the *IGHV* SHM% (SHM%) as a continuous parameter in *MYD88*-mutated and wild type groups.

TTT was defined as the time from initial diagnosis to first treatment. OS was defined as the time from diagnosis to death or the last follow-up. TTT and OS curves were generated using the Kaplan–Meier method and compared using the log-rank test. Cox proportional hazards regression was used for univariate and multivariate analyses. All statistical tests were two-sided, and a *p* value of <0.05 indicated statistical significance.

## Figures and Tables

**Figure 1 ijms-24-03177-f001:**
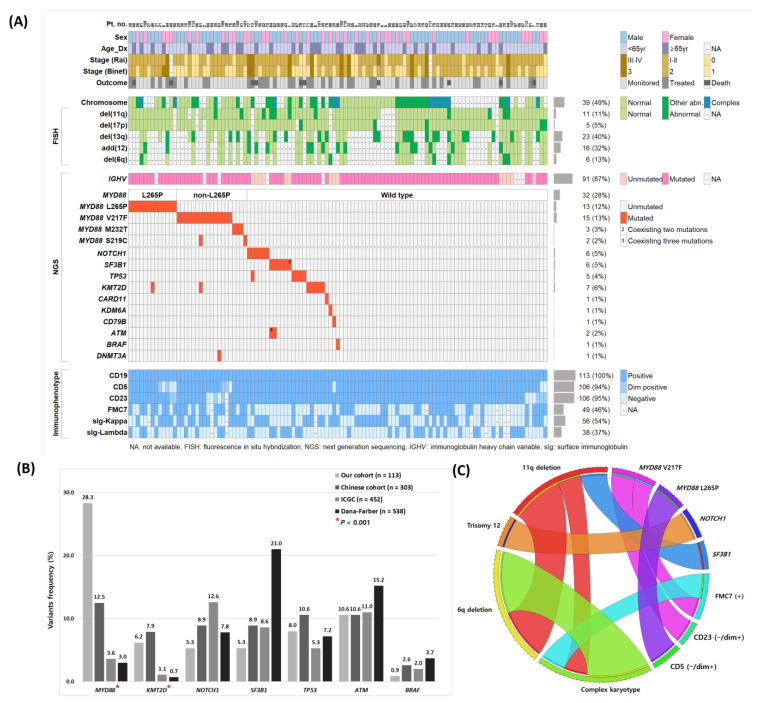
(**A**) Distribution and incidence of molecular, cytogenetic, and immunophenotypic parameters, and their clinical profiles. (**B**) Topography compares the frequency of genetic aberrations among our cohort (*n* = 113), Chinese (*n* = 303), ICGC (*n* = 452), and Dana-Farber (*n* = 538) cohorts. Red asterisks indicate *p* < 0.001. (**C**) The Circos plot indicates recurrent co-occurrences of genetic aberrations and immunophenotypes via ribbon widths. Abbreviations: ICGC, International Cancer Genome Consortium; NA, not available.

**Figure 2 ijms-24-03177-f002:**
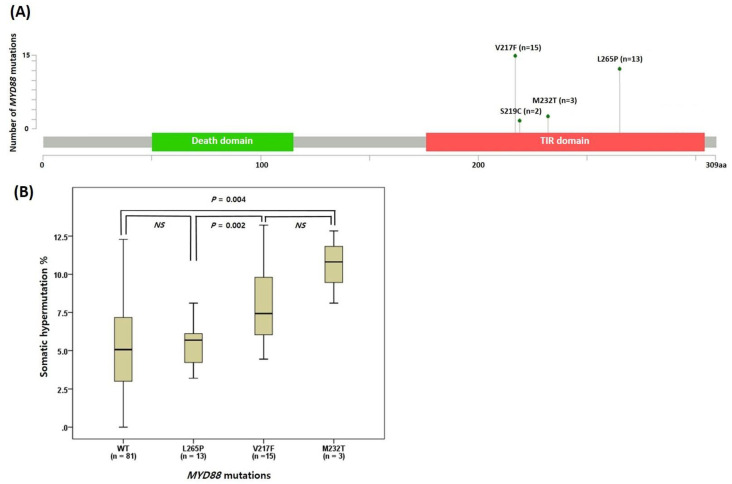
*MYD88* mutations in Korean CLL patients. (**A**) Location and frequency of the *MYD88* mutation types. (**B**) Somatic hypermutation percentage (V gene mutation ≥ 2%) according to *MYD88* mutation types. Abbreviations: aa, amino acid; TIR, Toll/interleukin-1 receptor; WT, wild type; NS, not significant.

**Figure 3 ijms-24-03177-f003:**
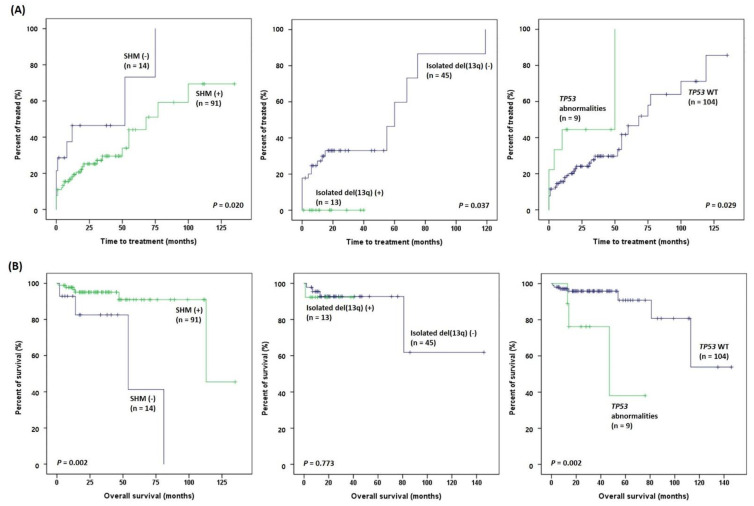
Survival analyses of clinical outcomes in patients with CLL according to somatic hypermutation (SHM) and genetic aberrations status. Kaplan–Meier estimates of (**A**) time to treatment (TTT) and (**B**) overall survival (OS).

**Figure 4 ijms-24-03177-f004:**
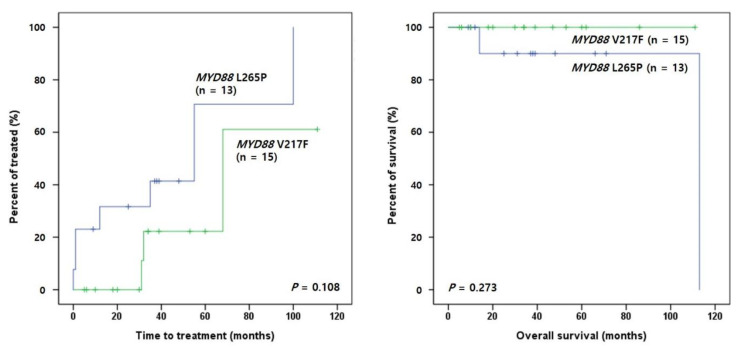
Survival analyses of clinical outcomes in patients with *MYD88*-mutated CLL. Kaplan–Meier estimates of time to treatment (TTT) and overall survival (OS).

**Figure 5 ijms-24-03177-f005:**
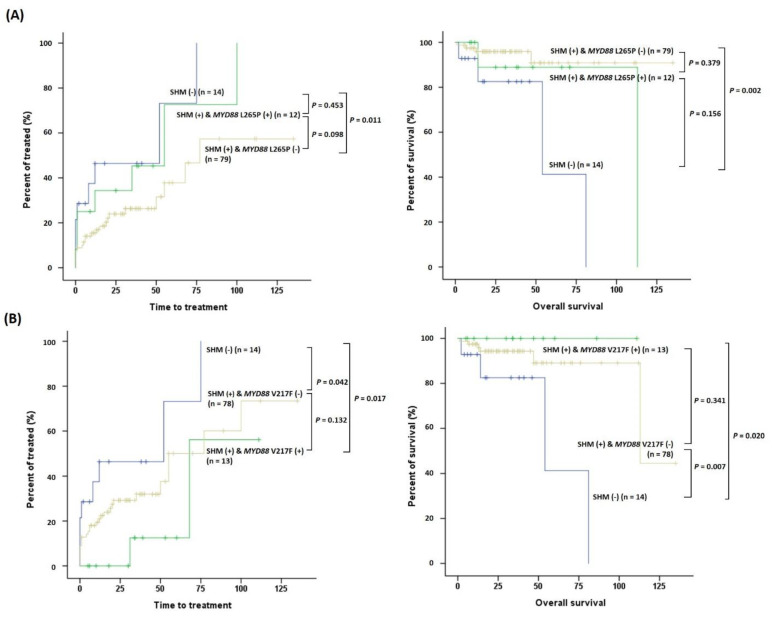
Survival analyses of clinical outcomes in patients with CLL according to somatic hypermutation (SHM) and *MYD88* mutation status. Subgroup analysis of (**A**) SHM and L265P status, and (**B**) SHM and V217F status. Kaplan–Meier estimates of time to treatment (TTT) and overall survival (OS).

**Table 1 ijms-24-03177-t001:** Clinical and biological characteristics of 113 patients with chronic lymphocytic leukemia (CLL) according to *MYD88* mutations. Asterisks indicate *p* < 0.05.

	Unmutated *MYD88* (*n* = 81)		Mutated *MYD88* (*n* = 32)		*p*
Characteristics	No.	%	No.	%	
Male sex	53	65.4%	21	65.6%	1.000
Age, years					0.243
Median	59		62		
Range	32–81		35–87		
Rai stage					0.208
0	10	12.5%	5	16.1%	
I–II	60	75.0%	25	80.6%	
III–IV	10	12.5%	1	3.2%	
Binet Stage					0.466
A	32	40.0%	14	45.2%	
B	39	48.8%	15	48.4%	
C	9	11.3%	2	6.5%	
Immunophenotype					
Atypical Immunophenotype	29/81	35.8%	26/32	81.3%	<0.001 *
CD5(−/dim+)	3/81	3.7%	8/32	25.0%	0.002 *
CD23(−/dim+)	6/81	7.4%	7/31	22.6%	0.043 *
FMC7(+)	28/77	36.4%	21/30	70.0%	0.002 *
Genetics					
Cytogenetic abnormalities	49/75	65.3%	12/32	37.5%	0.010 *
del(11q)	9/73	12.3%	3/29	10.3%	1.000
del(17p)	5/72	6.9%	0/29	0.0%	0.318
del(13q)	19/45	42.2%	4/13	30.8%	0.534
add(12)	15/41	36.6%	2/10	20.0%	0.463
del(6q)	6/37	16.2%	2/11	18.2%	1.000
Other mutations	27/81	34.2%	3/32	10.3%	
*KMT2D*	5/81	6.2%	2/32	6.3%	1.000
* SF3B1*	6/81	7.4%	0/32	0.0%	0.181
* NOTCH1*	6/81	7.4%	0/32	0.0%	0.181
* TP53*	5/81	6.2%	0/32	0.0%	0.319
* IGHV* (>2% mutation)	62/76	81.6%	29/29	100.0%	0.010 *

## Data Availability

The data presented in this study are available on request from the corresponding author. The data are not publicly available due to ethical concern.

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
