# Peer review of "Genetic and Clinical Characteristics of Korean Chronic Lymphocytic Leukemia Patients with High Frequencies of MYD88 Mutations"

_ijms, 2023, doi:10.3390/ijms24043177_

Round 1
Reviewer 1 Report
Where is the comparison with other populations shown in the manuscript? Without clear and comprehensive comparison of the observations made in the manuscript to similar observations made in other population cohorts, it is difficult to understand the significance of the study.
Is the chemotherapeutic regimen for CLL patients in Korea different from what is prescribed to CLL patients in other countries? How does the current study help explain treatment strategy and disease management in CLL patients in Korea?
Reviewer 2 Report
In the article “Genetic and Clinical Characteristics of Korean Chronic Lymphocytic Leukemia with a High MYD88 Mutation Frequency”, the authors investigated the profiles of genetic mutations and cytogenetic abnormalities in Korean patients with CLL, and associated the molecular features with clinical features, like the time to treatment requirement and prognosis. As pointed out by the authors, CLL is relatively rare in Asian races, and, therefore, the aim of this study to investigate if the genetic mutational profile is different between CLL patients in Western countries and those in Asian countries seemed to be reasonable. However, the results shown in this research revealed that genetic and cytogenetic profiles and their clinical impacts in Korean patients are quite similar to those observed and have been repeatedly reported in findings with CLL patients in Western countries. Unfortunately, there was no novel or sound finding which gives new insight into the molecular/genetic biology of CLL in this article, and there are a lot of missing data, especially about chromosome data, in patients analyzed in this study. Thus, the data presented here may have some importance or utility as the epidemiological data, something like a database, in the Korean hematology community, while I would argue about the scientific importance in the global scientific field, especially the hematological oncology field.
Round 2
Reviewer 1 Report
The authors have responded to the comments satisfactorily.
However, in the absence of new information about the biology of CLL, this work will only serve to act as a confirmatory report or at best an incremental finding, only applicable to the selected population cohort.
Author Response
We thank the reviewer for this comment. We carefully considered your comments and, we believe that our paper has its own originality for the following three reasons.
First, the effects of MYD88 mutations in CLL patients are still debatable. The presence of MYD88 has been linked to a good outcome in a specific group of younger patients with CLL [Blood 2014, 123, 3790-3796]. However, other researchers have found that MYD88 mutations are linked to a poor prognosis in CLL patients with SHM [Blood Cancer J 2017, 7, 651][Br J Haematol 2019, 184, 925-936]. In addition, some previous studies have also found no significant impact of MYD88 mutations on prognosis [Leukemia 2014, 28, 108-117][Blood 2015, 126, 1043-1044][Blood Cancer J 2020, 10, 86]. Therefore, the impact of MYD88 mutations on the prognosis of CLL patients is still a topic of debate.
Second, we included a subtype analysis of MYD88 mutations which has not been performed in other previous studies. Our data suggested an overall unfavorable effect of L265P and a favorable effect of V217F on CLL prognosis.
Finally, the number of MYD88-mutated CLL patients is sufficiently comparable to other studies. Our study included 32 patients with MYD88-mutated CLL including V217F (n=15) and L265P (n=13), with a considerably higher positive rate (28%) than others. For comparison, the abbreviated data below is attached.
MYD88-mutated CLL patients, positive rates, total patients
n=32, 28%, n=113, Our study
n=39, 13%, n=303, Yi et al. Leukemia 2021, 35, 2412-2415.
n=25, 8.8%, n=284, Qin et al. Blood Cancer J 2017, 7, 651.
n=12, 7.5%, n=160, Improgo et al. Br J Haematol 2019, 184, 925-936.
n=21, 2.0%, n=1039, Baliakas et al. Blood 2015, 126, 1043-1044.
n=55, 3.1%, n=1779, Shuai et al. Blood Cancer J 2020, 10, 86.
We hope that our responses satisfy you, and you now find our paper worthy of publication. We thank the valuable reviewer’s comments.
Reviewer 2 Report
I have no additional comment.
Author Response
We thank the valuable reviewer’s comments. We hope that the revised manuscript is now suitable for publication in your journal.